# Timing of Early Cholecystectomy for Acute Calculous Cholecystitis: A Multicentric Prospective Observational Study

**DOI:** 10.3390/healthcare11202752

**Published:** 2023-10-17

**Authors:** Paola Fugazzola, Fikri M. Abu-Zidan, Lorenzo Cobianchi, Francesca Dal Mas, Marco Ceresoli, Federico Coccolini, Simone Frassini, Matteo Tomasoni, Fausto Catena, Luca Ansaloni

**Affiliations:** 1Fondazione IRCCS Policlinico San Matteo, Division of General Surgery, 27100 Pavia, Italy; lorenzo.cobianchi@unipv.it (L.C.); simone.frassini01@universitadipavia.it (S.F.); matteotomasoni83@gmail.com (M.T.); l.ansaloni@smatteo.pv.it (L.A.); 2Department of Clinical, Diagnostic and Pediatric Sciences, University of Pavia, Via Alessandro Brambilla, 74, 27100 Pavia, Italy; 3The Research Office, College of Medicine and Health Sciences, United Arab Emirates University, Al-Ain 15551, United Arab Emirates; fabuzidan@uaeu.ac.ae; 4ITIR—Institute for Transformative Innovation Research, University of Pavia, 27100 Pavia, Italy; 5Department of Management, Ca’ Foscari University of Venice, 30123 Venice, Italy; francesca.dalmas@unive.it; 6General and Emergency Surgery, School of Medicine and Surgery, Milano-Bicocca University, 20900 Monza, Italy; marco.ceresoli@unimib.it; 7Department of Emergency and Trauma Surgery, Pisa University Hospital, University of Pisa, 56126 Pisa, Italy; federico.coccolini@gmail.com; 8General and Emergency Surgery, Bufalini Hospital, 47521 Cesena, Italy; faustocatena@gmail.com

**Keywords:** acute cholecystitis, cholecystectomy, surgery, timing, complication, mortality

## Abstract

The definition of Early Cholecystectomy (EC) is still debatable. This paper aims to find whether the timing of EC affects outcomes. The article reports a multicentric prospective observational study including patients with acute calculous cholecystitis (ACC) who had cholecystectomy within ten days from the onset of symptoms. Kruskall-Wallis test, Fisher’s Exact test, and Spearman rank correlation were used for statistical analysis. The patients were divided into three groups depending on the timing of the operation: 0–3 days, 4–7 days, or 8–10 days from the onset of symptoms. 1117 patients were studied over a year. The time from the onset of symptoms to EC did not affect the post-operative complications and mortality, the conversion, and the reintervention rate. The time represented a significant risk factor for intraoperative complications (0–3 days, 2.8%; 4–7 days, 5.6%; 8–10 days, 7.9%; *p* = 0.01) and subtotal cholecystectomies (0–3 days, 2.7%; 4–7 days, 5.6%; 8–10 days, 10.9%; *p* < 0.001). ACC is an evolutive inflammatory process and, as the days go by, the local and systemic inflammation increases, making surgery more complex and difficult with a higher risk of intraoperative complications. We recommend performing EC for ACC as soon as possible, within the first ten days of the onset of symptoms.

## 1. Introduction

10–15% of the general population is affected by cholelithiasis, and 20–40% of them will develop, in the course of their lives, secondary complications due to the presence of gallstones [1]. The first clinical presentation in 10–15% of patients with complications due to cholelithiasis is Acute Calculous Cholecystitis (ACC) [1,2].

The most widely used guidelines worldwide for the management of ACC are the Tokyo Guidelines (TGs) [2] and the World Society of Emergency Surgery (WSES) guidelines [1]. The TGs, published for the first time in 2007 and subsequently updated in 2013 and 2018, established for the first-time objective parameters for the diagnosis, classification and management of ACC [2]. The first edition of the WSES guidelines was published in 2016 and updated in 2020. In many respects, the 2018 TGs were in line with the recommendations of the 2020 WSES Guidelines, particularly about surgical indications for patients with severe ACC [1,2]. However, some differences between the TGs and the WSES Guidelines emerge, even on critical topics, e.g., in the timing of treatment.

Early cholecystectomy (EC) represents the gold standard for treating ACC [1,2]. It is superior to both intermediate cholecystectomy (performed between 7 days and 6 weeks of hospital admission) and delayed cholecystectomy (performed between 6 weeks and 3 months from hospital admission) [1]. EC has shorter total hospital stays and lower costs but longer operative times [3,4]. 

However, the definition of EC is still debatable without available definitive data. The WSES guidelines defined “early cholecystectomy” when performed within 7 days of hospital admission and within 10 days of the onset of symptoms [1]. The TGs defined EC when performed as soon as possible, preferably within 72 h from the onset of symptoms, but even after this time [2].

Multiple studies and meta-analyses comparing different timing of EC revealed no significant association between timing of operation and post-operative mortality or morbidity [4,5,6,7], but longer post-operative lengths of stay (LOS) in the group of patients with longer times from admission to surgery [5,8]. On the other hand, a large retrospective observational study [9] including 43,870 patients in England who underwent emergency cholecystectomy on index admission, showed a significantly lower biliary complication rate in patients undergoing cholecystectomy within 3 days of admission. Another retrospective study [10], including 34,151 cholecystectomies for ACC showed that operations performed on hospital days 3–7 had increased 30-day mortality and morbidity in comparison to hospital day 1 or hospital day 2. On multivariable analysis, the number of days from admission to EC was an independent predictor of mortality.

Focusing on the conversion rate, some observational studies [5] showed that patients who underwent an operation later in the course of admission were more likely to require an open procedure. Other studies [8,11] did not find any differences in the conversion rate, but an increasing rate of difficult surgical procedures and an increasing operative time. A recent meta-analysis has found that cholecystectomy which was performed within 24 h of admission has not reduced the post-operative complications [7], but has reduced LOS. Many of these studies are not comparable because they consider different time intervals. A recent meta-analysis has shown that cholecystectomies performed within 72 h of symptoms have reduced conversion rate and LOS in comparison to cholecystectomy ≤ 7 days [4,7]. No differences in complication rate and bile duct injuries have been found [4]. The literature features three randomized controlled trials [12,13,14] and other prospective non-randomized studies [6,11] that compare different timings of EC from the onset of symptoms. Three randomized controlled trials randomized patients with ACC to receive EC within 72 h or after 4–7 days from the onset of symptoms. In these studies, no differences in post-operative complication rate and conversion rate were found, but patients who received EC within 72 h had a significantly shorter post-operative LOS. In the studies by Chandler et al. [12] and by Onuk et al. [13] there was no difference in the duration of the surgery, while in the study by Jan et al. [14] the operative time was longer in patients who were operated on after 72 h. Furthermore, Jan et al. [14] did not find a significant difference in intraoperative complications, while Chandler et al. [12] found significantly greater blood loss in those operated on after 72 h. However, none of the randomized controlled trials and the prospective non-randomized studies had enough powered sample size. The only study that reported data about sample size calculation [13], had a power of 34%. A Cochrane systematic review [15] highlighted the difficulty of obtaining sufficient data on this topic through randomized controlled trials because studies with enough power would involve thousands of patients. 

Accordingly, there is a lack of high-quality and properly powered studies that stratify the intra and post-operative risks of EC based on the delay of surgery from the onset of symptoms, especially when considering an inclusive time of 10 days from the onset of symptoms. The validation and comparison of Scores for Prediction of Risk for post-operative major Morbidity after cholecystectomy in Acute Calculous Cholecystitis (S.P.Ri.M.A.C.C.) study was conceived as a WSES prospective multicentre observational study on patients with ACC who are candidates for EC aiming to validate different scores in predicting post-operative complications [16]. This current paper is a posthoc analysis of the S.P.Ri.M.A.C.C. study aiming to define the effects of different timings of EC (within 10 days from the onset of symptoms) on intra and post-operative outcomes.

## 2. Materials and Methods

### 2.1. Ethical Considerations

The medical ethics board of the trial coordinating centre IRCCS San Matteo University Hospital, Pavia, Italy, approved the S.P.Ri.M.A.C.C. study protocol. All regional ethics committees of the participating centres provided secondary approval. Before enrollment, patients provided both verbal and written informed consent. The S.P.Ri.M.A.C.C. trial was carried out in line with the Helsinki Declaration.

### 2.2. Design

The S.P.Ri.M.A.C.C. study is an observational multicenter prospective study endorsed by the WSES. 1253 patients from 79 locations in 19 nations were enrolled between 1 September 2021, and 1 September 2022. The study was listed in LegalTrial.gov under case number NCT04995380. Patients were recruited in the preoperative period by the surgeons working in the centers that joined the study. Patients were enrolled after the examination of their condition and instrumental and biochemical investigations that allowed physicians to diagnose ACC. The full S.P.Ri.M.A.C.C. study protocol can be accessed via the study website https://sprimaccstudy.wixsite.com/website (accessed on 11 October 2023). The endpoint of the S.P.Ri.M.A.C.C. study was to prospectively validate and compare the performance of pre-operative risk prediction models (the Chole-Risk score, the POSSUM Physiological Score (PS), the modified Frailty Index, the Charlson Comorbidity Index, the American Society of Anesthesiologists-Performance Status (ASA-PS), the APACHE II score, the severity grade of ACC) in predicting in-hospital mortality, 30-day-mortality, in-hospital major morbidity (intended as Clavien-Dindo ≥ 3 complications) and 30-day-major morbidity in patients with ACC undergoing EC. 

The present work is a post-hoc analysis of data collected or S.P.Ri.M.A.C.C. study with the aim of defining the best timing for EC. 1117 participants were included in the study after patients with incomplete information regarding the timing of EC were excluded. Within 10 days after the onset of symptoms, EC was administered to all patients. The goal of the present study was to determine whether there was a statistically significant difference in the rate of intraoperative complications between individuals who underwent surgery at various times after the onset of their symptoms (0–3 days, 4–7 days, 8–10 days).

### 2.3. Studied Variables 

The intraoperative complication rate was the primary objective. It included haemorrhage above 500 mL, biliary tree injuries, bowel perforation, major vascular injuries, general anaesthetic respiratory complications, and general anaesthesia cardiac issues. The secondary endpoints were represented by the rate of conversion from laparoscopic to laparotomy, the rate of needing bailout procedures (subtotal cholecystectomy, fundus-first cholecystectomy, laparoscopic drainage only), the operative time, the in-hospital post-operative major complications (defined as complications with Common Terminology Criteria for Adverse Event, CTCAE ≥ 3), the 30-day post-operative major complications (CTCAE ≥ 3), the total LOS, the need for surgical or endoscopic reintervention or interventional radiology after cholecystectomy, the in-hospital mortality, and the 30-day mortality. 

### 2.4. Inclusion and Exclusion Criteria

Inclusion criteria were (1) to have a diagnosis of ACC as defined by 2018 TGs criteria, (2) to be a candidate for EC during the index admission (any surgical technique, laparoscopic or open procedures, including also bailout procedures such as subtotal cholecystectomy), (3) to be ≥18 years old, (4) to be stratified for the risk of common bile duct stones, and, in case of confirmation, receive preoperative ERCP, (5) to provide a signed and dated informed consent form and (6) to be willing to comply with all study procedures and be available for the duration of the study.

Exclusion criteria were (1) pregnancy or lactation, (2) ACC not related to a gallstone aetiology, (3) onset of symptoms > 10 days before cholecystectomy (patients with ACC associated with common bile duct stones who underwent pre-operative ERCP could have been included if they had received EC within 10 days from onset of symptoms), (4) concomitant cholangitis or pancreatitis, (5) intraoperative treatment of common bile duct stones, or (6) anything that would increase the risk for the patient or preclude the individual’s full compliance with or completion of the study.

### 2.5. Statistical Analysis

*Sample size*: for the sample size calculation, the investigators grounded on the randomized controlled trial by Jan et al. [14] in which patients with ACC were randomized to receive EC within 72 h from the onset of symptoms (group 1) or after 72 h up to 7 days from the onset of symptoms (group 2). In the study, the intraoperative complication rate in group 1 was 2%, while in group 2 was 6%. 1000 patients are required to have a 90% chance of detecting, as significant at the 5% level, an increase in the primary outcome measure. The patients were divided into three different groups depending on the time from onset of symptoms: group I from 0–3 days, Group II from 4–7 days, and Group III from 8–10 days.

*Variables comparison*: The three groups were compared using the Kruskall-Wallis test for continuous or ordinal data and Fisher’s Exact test for categorical data. Spearman rank correlation was used to study the correlation between the continuous or ordinal data. A P value of less than 0.05 was accepted as significant. SPSS version 26 was used for comparison.

## 3. Results

58.7% of patients received EC within 3 days, 32.2% from 4 to 7 days, and 9.0% from 8 to 10 days from the onset of symptoms. The mean age was 59 and the mean POSSUM (Physiological and Operative Severity Score for the enumeration of Mortality and morbidity) Physiological score (PS) was 20.7. The intraoperative complication rate was 4.1%. The most frequent complication was intraoperative bleeding > 500 mL (16 patients). Other intraoperative complications were biliary tree injuries (8 patients), bowel perforation (1 patient), respiratory complications (3 patients), cardiac complications (3 patients), and others (16 patients). 8.4% of patients needed a bail-out procedure (49 subtotal cholecystectomies, 65 patients treated with a fundus-first technique, and 1 patient received only laparoscopic drainage). Preoperative characteristics of the three groups of patients are reported in Table 1. Patients were similar in terms of age and BMI. Patients operated after 8–10 days from the onset had a significantly higher POSSUM PS (median (IQR): 0–3 days, 19 (15–24); 4–7 days, 19 (16–24); 8–10 days, 21 (17–26); *p* = 0.012). Patients operated on within 3 days from onset of symptoms had lower ACC severity grades (mean (SD): 1.6 (0.5) for 0–3 days, 1.8 (0.4) for 4–7 days, 1.7 (0.5) for 8–10 days; *p* = 0.012).

Table 2 shows intraoperative outcomes. A higher number of days from the onset of symptoms to EC was a significant risk factor for longer operative times (median (IQR): 0–3 days, 90 (60–120) minutes; 4–7 days, 100 (65–134.5) minutes; 8–10 days, 107 (74–145) minutes; *p* < 0.001), for needing of bail-out procedures (0–3 days, 6.9%; 4–7 days, 9.7%; 8–10 days, 13.9%; *p* = 0.037) and for intraoperative complications (0–3 days, 2.8%; 4–7 days, 5.6%; 8–10 days, 7.9%; *p* = 0.01). Analyzing the kind of bail-out procedures, the rate of subtotal cholecystectomies significantly increased with the increase of days from onset (0–3 days, 2.7%; 4–7 days, 5.6%; 8–10 days, 10.9%; *p* < 0.001). Figure 1 shows the box-and-whisker plot of the operative time by the time between the onset of symptoms and surgery. The time interval from the onset to EC did not affect the conversion rate to open surgery.

Table 3 shows post-operative outcomes. A higher number of days from the onset of symptoms to EC was a significant risk factor for a longer LOS (median (IQR) LOS: 0–3 days, 3 (2–5) days; 4–7 days, 5 (3–7) days; 8–10 days, 8 (3–11) days; *p* < 0.001; patients with a LOS longer than 10 days: 0–3 days, 6.67%; 4–7 days, 10.6%; 8–10 days, 24.8%; *p* < 0.001) (Figure 2). The time from the onset to EC did not affect the reintervention rate, the postoperative complications, and mortality.

There was a statistically significant, small positive correlation between days from onset of symptoms and LOS (r = 0.26, *p* < 0.001), POSSUM PS (r = 0.01, *p* < 0.001) and operative time (r = 0.14, *p* < 0.001) (Table 4).

## 4. Discussion

To our knowledge, the present investigation represents the first prospective study focusing on the ideal timing of EC with a sample size calculated on a high study power. These data showed that the risk of intraoperative complications, the risk of needing a bail-out procedure (e.g., subtotal cholecystectomy), the operative time, and, to a lesser degree, the LOS increase for each time interval (0–3 days, 4–7 days, 8–10 days) from the onset of symptoms to EC. The evolution of the inflammatory process, with the passage of days, makes it more difficult to dissect tissues, recognize structures, and increase the tendency to bleed. The ACC severity grade and the POSSUM PS rise with the passing of days due to the local and systemic inflammation exacerbation. 

These findings could be explained by the pathogenesis of ACC. The first 2–4 days of ACC are the phase of edematous cholecystitis, during which congestion and oedema are evident findings. Then, at 3–5 days, ACC progresses in the necrotizing phase, characterized by bleeding and necrosis. From 7–10 days, the disease progresses to its purulent phase, also known as suppurative cholecystitis. If the disease is left untreated, it progresses to subacute cholecystitis and it eventually becomes chronic cholecystitis [17].

So, the pathogenesis of ACC is primarily inflammatory due to obstruction of biliary outflow that progresses into a circulatory disorder. For this reason, ACC has similarities with bowel ileus (particularly strangulated ileus), in which mechanical obstruction occurs first, and circulatory disturbance follows afterward, eventually resulting in inflammation and tissue necrosis of the obstructed tract. In summary, ACC is caused by a mechanical stimulus, secondary bacterial infection and bile irritation follows afterward, causing inflammation to progress [17,18].

Despite the greater difficulty of the intervention, the delay in performing EC, if within 10 days from the onset of symptoms, did not affect post-operative complications, post-operative mortality, risk of conversion to open surgery, and need for reintervention.

The definition of EC in terms of timing is still debatable [4,5,6,7,8,9,10,11]. Most of the existing studies are unpowered or they are not comparable because they consider different timing definitions. Some consider the time interval from the onset of symptoms while others from hospitalization. Patients can go to the emergency department after very different intervals of time from the onset of symptoms, and then at different stages of the disease. This may depend on age, comorbidities, geographic location, pain tolerance and social class. This can lead to poor population uniformity when the timing is based on the time of admission. Basing the timing on the onset of symptoms allows to create more homogeneous patient groups for the stage of disease and the surgical risk. 

Compared with other surgical urgencies, physicians often procrastinate cholecystectomy for ACC. Hospitals and community care often fail to ensure patients with ACC an ideal timing for surgery. General practitioners, emergency physicians, and surgeons often try to treat ACC medically before considering surgery. This, combined with the organizational issues of operating rooms, increases the time between the onset of symptoms and EC. It worsens the general and local inflammatory condition, increases the surgical complexity, and increases the risk of intraoperative complications.

The time between the onset of symptoms and the presentation of the patient in the emergency department does not depend on the physician. However, the physician, considering the organizational issues within the health care system and the availability of the operating rooms, should do everything possible to ensure a patient with ACC has a cholecystectomy performed as early as possible.

Probably, a proper surgeon-patient and surgeon-general practitioner knowledge translation [19] could shorten the time between the onset of symptoms and emergency department presentation. Surgeons should recommend symptomatic patients with cholelithiasis during outpatient visits to go to the hospital as soon as possible. Furthermore, the time between the arrival in the emergency room and the surgical visit should be optimized. Clinical examination, blood tests, and ultrasound are usually sufficient to diagnose ACC. Emergency doctors should not delay the management by performing unneeded examinations or attempts for medical therapy when there is a clear indication for surgery. 

Main bile duct stones associated with ACC often delay EC because of the need for MRCP or ERCP to have a clear management plan. Probably, a single-stage intraoperative ERCP or a laparoscopic cholangioscopy combined with laparoscopic cholecystectomy will reduce the time for EC [20]. 

Our study has some limitations. It is a non-randomized study with possible confounding factors. However, given the large sample size required, it would be difficult to carry out a randomized controlled trial on this topic. Furthermore, we were interested in the generalizability of the study, which will contain more heterogenicity in the data. This included some developing countries with a lack of training in research methodology, including performing randomized controlled trials. To our knowledge, the prospective nature and the large sample size of our study, despite being type II research data, provide the highest quality data available in the literature. An adequate randomized controlled trial would be possible only through a multicenter study with a high degree of organization and international cooperation.

## 5. Conclusions

In conclusion, considering that EC is superior to delayed and interval cholecystectomy [1,3,4], this study clarifies the best timing of EC. Our study has shown that delaying EC up to ten days from the onset of symptoms does not affect post-operative complications and mortality. However, ACC is an evolutive inflammatory process, and, as the days go by, the local and systemic inflammation increases which makes surgery more complex and difficult with a higher risk of intraoperative complications. We recommend performing EC for ACC as soon as possible within the first ten days of the onset of symptoms. 

## Figures and Tables

**Figure 1 healthcare-11-02752-f001:**
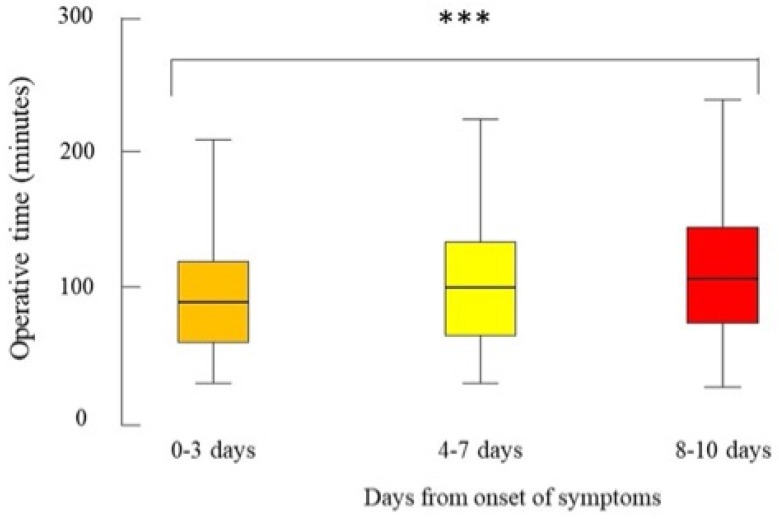
Box-and-whisker plot of the operative time for the patients who were operated on for acute cholecystitis from September 2021 to September 2022 from 79 centres in 19 countries by the time between the onset of symptoms and surgery, n = 117. The box represents the 25th to the 75th percentile IQR. The horizontal line within each box represents the median. *** *p* < 0.001, Kruskal-Wallis test.

**Figure 2 healthcare-11-02752-f002:**
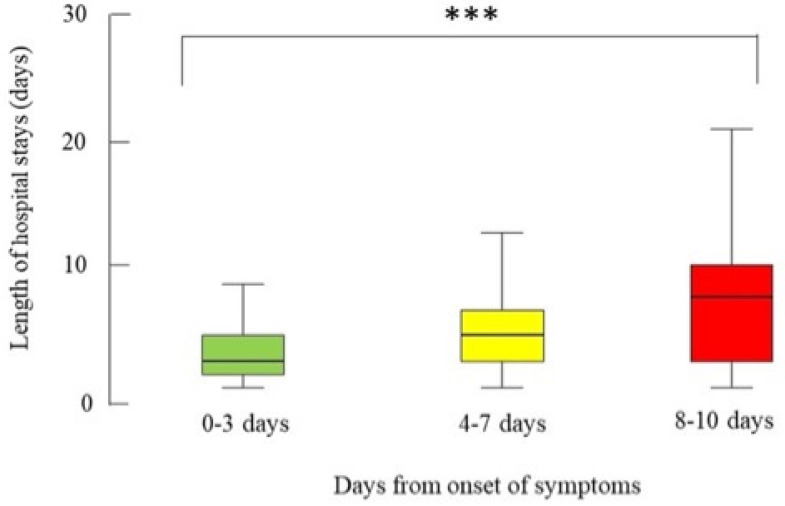
Box-and-whisker plot of the length of hospital stay (days) of the patients who were operated on for acute cholecystitis from September 2021 to September 2022 from 79 centres in 19 countries by the time between the onset of symptoms and surgery, n = 117. The box represents the 25th to the 75th percentile IQR. The horizontal line within each box represents the median. *** *p* < 0.001, Kruskal-Wallis test.

**Table 1 healthcare-11-02752-t001:** Preoperative characteristics of the patients who were operated on for acute cholecystitis from September 2021 to September 2022 from 79 centres in 19 countries by the time between the onset of symptoms and surgery, n = 117.

	Onset of Symptoms			
Variables	0–3 Days n = 656	4–7 Days n = 360	8–10 Days n = 101	*p* Value
Age (years)	59 (46–72)	62 (50–74)	61 (46–74)	0.14
BMI	26.7 (24.3–29.4)	26.7 (24.2–29.4)	26.3 (23.6–28.9)	0.4
ACC severity grade	2 (1–2) *	2 (2–2) *	2 (1–2) *	<0.001
POSSUM physiological score	19 (15–24)	19 (16–24)	21(17–26)	0.012
Days from admission to surgery	1 (0–1)	2 (0–3)	6 (2–8)	<0.001

BMI: Body Mass Index, ACC: Acure Cholecystitis, POSSUM: Physiological and Operative Severity Score for the enumeration of Mortality and morbidity, PS: physiological score. Data are presented as median (IQR). *p* value Kruskall-Wallis test. * mean (SD) of ACC severity grade: 1.6 (0.5) for 0–3 days, 1.8 (0.4) for 4–7 days, 1.7 (0.5) for 8–10 days.

**Table 2 healthcare-11-02752-t002:** Intraoperative outcome of the patients who were operated on for acute cholecystitis from September 2021 to September 2022 from 79 centres in 19 countries by the time between the onset of symptoms and surgery, n = 117.

	Onset of Symptoms			
Variables	0–3 Days n = 656	4–7 Days n = 360	8–10 Days n = 101	*p* Value
Operative time (minutes)	90 (60–120)	100 (65–134.5)	107 (74–145)	<0.001
Conversion to open surgery	48 (7.9%)	32 (9.9%)	9 (9.5%)	0.54
Bail-out procedure: Subtotal cholecystectomy Fundus-first technique Drainage only	45 (6.9%) 18 (2.7%) 34 (5.2%) 1 (0.2%)	35 (9.7%) 20 (5.6%) 24 (6.7%) 0 (0.0%)	14 (13.9%) 11 (10.9%) 7 (6.9%) 0 (0%)	0.037 <0.001 0.52 0.99
Intraoperative complications	18 (2.8%)	20 (5.6%)	8 (7.9%)	0.01

Data are presented as median (IQR) or number (%). Percentages were calculated from valid available data. *p* value Kruskall-Wallis test of Fisher’s Exact test as appropriate.

**Table 3 healthcare-11-02752-t003:** Post-operative outcomes of the patients who were operated on for acute cholecystitis from September 2021 to September 2022 from 79 centres in 19 countries by the time between the onset of symptoms and surgery, n = 117.

	Onset of Symptoms			
Variables	0–3 Days n = 656	4–7 Days n = 360	8–10 Days n = 101	*p* Value
Reintervention	16 (2.4%)	6 (1.7%)	0 (0%)	0.29
Inhospital major complication	38 (5.8%)	15 (4.2%)	6 (5.9%)	0.49
30-day major complications	48 (7.3%)	22 (6.2%)	7 (7.1%)	0.78
Inhospital mortality	5 (0.8%)	5 (1.4%)	1 (1%)	0.52
30-day mortality	5 (0.8%)	7 (1.9%)	1 (1%)	0.25
LOS > 10 days	43 (6.6%)	38 (10.6%)	25 (24.8)	<0.001
LOS (days)	3 (2–5)	5 (3–7)	8 (3–11)	<0.001

LOS: length of stay. Data are presented as median (IQR) or number (%). *p* value Kruskall-Wallis test of Fisher’s Exact test as appropriate.

**Table 4 healthcare-11-02752-t004:** Correlations between different continuous and ordinal data of the patients who were operated on for acute cholecystitis from September 2021 to September 2022 from 79 centres in 19 countries by the time between the onset of symptoms and surgery, n = 117.

		Length of Stay	Days from Admission to EC	POSSUM PS	Operative Time
Days from onset to EC	Correlation	0.26	0.53	0.01	0.14
	*p* value	<0.001	<0.001	0.002	<0.001
Length of stay	Correlation	-------	0.35	0.31	0.33
	*p* value	-------	<0.001	<0.001	<0.001
Days from admission	Correlation	-------	-------	0.06	0.167
	*p* value	-------	-------	0.049	<0.001
POSSUM score	Correlation	-------	-------	-------	0.16
	*p* value	-------	-------	-------	<0.001

EC: early cholecystectomy, POSSUM: Physiological and Operative Severity Score for the enumeration of Mortality and morbidity, PS: physiological score. *p* value = Pearson rank correlation.

## Data Availability

The data presented in this study are available on request from the corresponding author.

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
