# Peer review of "Timing of Early Cholecystectomy for Acute Calculous Cholecystitis: A Multicentric Prospective Observational Study"

_healthcare, 2023, doi:10.3390/healthcare11202752_

Round 1
Reviewer 1 Report
A very good meta-analysis regarding the timing for acute cholecystitis surgery.
The risk of intraoperative complications is higher when delaying the surgery.
It is recommended to perform EC for ACC as soon as possible.
The main question addressed by the research:
Have you identified the percentage of conversion in acute cholecystitis? Is the convesrion infulenced by the delay? The topic of Acute cholecystitis is relevant in the surgical field, being a surgical emergency and it adress a specific gap in the field. The subject of the paper, regarding the timing of surgery in acute cholecystitis, reaffirm the importance of quick surgical treatment, based on a large group of study. The study has some limitations. It is a non-randomized study with possible confounding factors.In the future studies you will need more heterogenity in the data. The conclusions are consistent with the evidence and arguments presented and they do address the main question posed. The references are appropriate.Author Response
Dear reviewer, thank you very much for your comment.
The percentage of conversion is reported in Table 2, but it is not significantly influenced by the delay.
Reviewer 2 Report
The submitted paper is original article entitled as Timing of early cholecystectomy: A multicentric prospective observational study.
Congratulations on your great work.
However, there are some points to be revised.
1. The title is somewhat vague. The indication of early cholecystectomy should be stated clearly. I think acute cholecystitis should be added in the title.
2. This study has strength as a prospective study. However, excellent prospective study should define everything very strictly and therefore it can minimize the bias.
Are there any standards for operation method, use of antibiotics, participating surgeon’s experience, and so on?
For prospective study, flow diagram of patients enrolled is needed. For example, how many patients were screened, how many patients were excluded due to not agreeing with informed consent, incomplete information, or insertion of PTGBD, etc……
3. Severity of acute cholecystitis is one of the most important factors associated with postoperative morbidity. According to severity, additional subgroup analysis should be performed.
4. In postoperative outcomes, LOS means postoperative length of stay? If you mean whole LOS after admission, it should be postoperative length of stay.
5. In table 3, what do you mean reintervention? There is no definition in the method.
6. What did you mean major complication? There is no definition in the method. Clavien-Dindo classification is very useful to show severity of complications.
7. What is bail out procedures?
8. Your study demonstrated that cholecystectomy becomes more difficult as the days go by, with higher risk of intraoperative complications. And 3 days form the onset of symptom group showed the best outcome. However, the conclusion and your recommendation is to perform EC within the first ten days of the onset of symptoms. 10 day s from the onset of symptom is only your criteria. Therefore, I think you should not limit the timing within 10 days.
English should be corrected by native speaker
Author Response
- The title is somewhat vague. The indication of early cholecystectomy should be stated clearly. I think acute cholecystitis should be added in the title.
Thank you very much for the comment. We have modified the title in “Timing of early cholecystectomy for Acute Calculous Cholecystitis: A multicentric prospective observational study”
- This study has strength as a prospective study. However, excellent prospective study should define everything very strictly and therefore it can minimize the bias.
Are there any standards for operation method, use of antibiotics, participating surgeon’s experience, and so on?
For prospective study, flow diagram of patients enrolled is needed. For example, how many patients were screened, how many patients were excluded due to not agreeing with informed consent, incomplete information, or insertion of PTGBD, etc……
Thank you very much for the comment. In the method section we have added the link where the complete SPRIMACC protocol could be found. As reported in the manuscript, from the 1253 patients in SPRIMACC study, “1117 participants were included in the present study after patients with incomplete information regarding the timing of EC were excluded”
- Severity of acute cholecystitis is one of the most important factors associated with postoperative morbidity. According to severity, additional subgroup analysis should be performed.
Thank you very much for the comment. In table 1 we have reported the severity of ACC of included patients, divided in the three periods from onset of symptoms. However, the results of SPRIMACC study (Fugazzola P, Cobianchi L, Di Martino M, Tomasoni M, Dal Mas F, Abu-Zidan FM, Agnoletti V, Ceresoli M, Coccolini F, Di Saverio S, Dominioni T, Farè CN, Frassini S, Gambini G, Leppäniemi A, Maestri M, Martín-Pérez E, Moore EE, Musella V, Peitzman AB, de la Hoz Rodríguez Á, Sargenti B, Sartelli M, Viganò J, Anderloni A, Biffl W, Catena F, Ansaloni L; S.P.Ri.M.A.C.C. Collaborative Group. Prediction of morbidity and mortality after early cholecystectomy for acute calculous cholecystitis: results of the S.P.Ri.M.A.C.C. study. World J Emerg Surg. 2023 Mar 18;18(1):20. doi: 10.1186/s13017-023-00488-6. PMID: 36934276; PMCID: PMC10024826) showed that ACC severity grade have a lower accuracy in predicting postoperative morbidity than POSSUM PS in these patients.
- In postoperative outcomes, LOS means postoperative length of stay? If you mean whole LOS after admission, it should be postoperative length of stay.
Thank you for the comment. LOS means total length of stay (from admission to discharge). Unfortunately, because this paper is a post-hoc analysis of data included in SPRIMACC study, the postoperative length of stay is not available. However, because total LOS is strictly related to costs incurred and resourced used by the healthcare system, we think that it could be an important outcome to consider. We have specified the mean of LOS in methods.
- In table 3, what do you mean reintervention? There is no definition in the method.
Thank you very much, we have added an explanation in methods
- What did you mean major complication? There is no definition in the method. Clavien-Dindo classification is very useful to show severity of complications.
Thank you very much for the comment. In the method section it is reported “major complications (defined as complications with Common Terminology Criteria for Adverse Event, CTCAE >=3)”
- What is bail out procedures?
Thank you very much for the comment. In the method section it is reported “bailout procedures (subtotal cholecystectomy, fundus-first cholecystectomy, laparoscopic drainage only)”
- Your study demonstrated that cholecystectomy becomes more difficult as the days go by, with higher risk of intraoperative complications. And 3 days form the onset of symptom group showed the best outcome. However, the conclusion and your recommendation is to perform EC within the first ten days of the onset of symptoms. 10 day s from the onset of symptom is only your criteria. Therefore, I think you should not limit the timing within 10 days.
Thank you very much. Present study found that, despite the greater difficulty of the intervention, the delay in performing EC, if within 10 days from the onset of symptoms, did not affect post-operative complications, postoperative mortality, risk of conversion to open surgery and need for reintervention. For these reasons we recommend performing EC for ACC as soon as possible to reduce surgical difficulty, however it could be performed within the first ten days of the onset of symptoms without affecting complications and mortality.
Comments on the Quality of English Language. English should be corrected by native speaker.
Thank you very much, a native English speaker have corrected the manuscript.
Reviewer 3 Report
The article titled "Effect of Timing on Early Cholecystectomy for Acute Calculous Cholecystitis: A Prospective Multicenter Study" examines the impact of different timings for early cholecystectomy (EC) on intra and postoperative outcomes for patients with acute calculous cholecystitis (ACC). The study aims to provide insights into the optimal timing for EC in ACC patients. Below is a review of the article:
-
Clarity of Research Objectives: The article clearly states the research objective to determine the effects of different timings of early cholecystectomy on intra and postoperative outcomes in ACC patients. This objective is well-defined and provides a clear focus for the study.
-
Relevance and Significance: The study addresses an important clinical question regarding the timing of early cholecystectomy in ACC, which has practical implications for patient management. The authors provide relevant background information and cite existing guidelines, establishing the significance of their research.
-
Methodology:
- Ethical Considerations: The study outlines the ethical considerations, including approval from the medical ethics board and informed consent from patients, ensuring that ethical standards were adhered to.
- Design and Sample Size: The study employs a prospective multicenter design with a large sample size. The rationale for the sample size calculation is well-justified.
- Inclusion and Exclusion Criteria: The inclusion and exclusion criteria are clearly defined, contributing to the study's internal validity.
- Statistical Analysis: The statistical analysis is appropriate, and the statistical tests suit the data. The correlations presented enhance the interpretation of the results.
-
Presentation of Results:
- The presentation of results is clear and well-organized, with tables and figures aiding in understanding the data.
- The authors effectively report vital findings related to intra and postoperative outcomes, highlighting the impact of timing on various parameters.
-
Discussion and Interpretation:
- The discussion section provides a comprehensive analysis of the results and their implications.
- The authors discuss the study's limitations, including its non-randomized nature, and provide a reasonable explanation for the absence of a randomized controlled trial (RCT) due to practical challenges.
- The conclusion is logically derived from the study findings, emphasizing the importance of performing EC within ten days of symptom onset.
-
Quality of Evidence:
- The study's prospective multicenter design and large sample size contribute to the robustness of the evidence.
- The authors appropriately acknowledge the limitations of their study, which adds transparency and credibility to their findings.
-
Clarity and Writing Style:
- The article is generally well-written and organized.
- The use of tables and figures aids in data presentation.
- However, there are some minor typographical errors, such as missing spaces, that should be addressed.
-
Citation and References:
- The citation style adheres to academic standards.
- The article mentions that citations will be added by editorial staff during production, which is a minor issue that should be resolved.
In summary, the article presents a well-structured and valuable study on the timing of early cholecystectomy in acute calculous cholecystitis. It addresses an important clinical question, employs a robust methodology, and provides clear and meaningful results. The discussion and interpretation of findings are sound, and the limitations are appropriately acknowledged. Minor proofreading for typographical errors is recommended. Overall, the study contributes significantly to understanding the optimal timing for early cholecystectomy in ACC patients.
Author Response
Dear Reviewer, thank you very much for your comments. We have addressed the minor typographical errors.
Round 2
Reviewer 3 Report
This article reached the quality of our journal.